# Congenital Cytomegalovirus Screening in Massachusetts Birth Hospitals: A Statewide Survey

**DOI:** 10.3390/ijns8040065

**Published:** 2022-12-13

**Authors:** Cheryl K. Glovsky, Kendall Carroll, Naomi Clark, Peter Colleran, Vanessa Colleran, Shayne Gaffney, Margaret Kenna, Evelyn Kuhns-Rankin, Tracy Evans Luiselli, Talia Mango, Barbara Morris, Charlotte Mullen, Matthew Stenerson, Laura Gibson, Michael S. Cohen

**Affiliations:** 1Department of Audiology, Massachusetts Eye and Ear, Boston, MA 02114, USA; 2Massachusetts CMV Coalition, Boston, MA 02115, USA; 3Department of Otolaryngology and Communication Enhancement, Children’s Hospital Boston, Boston, MA 02115, USA; 4Department of Otolaryngology-Head and Neck Surgery, Harvard Medical School, Boston, MA 02115, USA; 5New England Consortium on Deafblindness, Watertown, MA 02472, USA; 6Department of Otolaryngology-Head and Neck Surgery, Massachusetts Eye and Ear, Boston, MA 02114, USA; 7Infectious Diseases and Immunology, Department of Medicine and Pediatrics, UMass Memorial Medical Center, Worcester, MA 01605, USA

**Keywords:** congenital cytomegalovirus, newborn screening, congenital hearing loss

## Abstract

This study sought to assess the current state of screening for congenital cytomegalovirus infection in newborns among birth hospitals and newborn nurseries in the state of Massachusetts. A survey assessing hospital protocols for cytomegalovirus testing in newborns was distributed to all birth hospitals and newborn nurseries in Massachusetts from November 2020 to February 2021. 73.3% of hospitals responded to at least one survey question. Of these, fewer than half (48.5%) had any established approach for neonatal cytomegalovirus screening. Salivary polymerase chain reaction was the most common testing modality. Most hospitals did not perform confirmatory testing for positive test results. Most respondents (87.9%) did not know or did not answer how results of cCMV screening were reported to families and who was responsible for coordinating care for cCMV-infected infants. We conclude that congenital cytomegalovirus screening protocols are absent or incomplete in most Massachusetts birth hospitals and newborn nurseries. A cohesive strategy involving standardized education and screening guidelines is needed to reduce the incidence and burden of congenital cytomegalovirus disease on children and their families.

## 1. Introduction

Congenital cytomegalovirus (cCMV) is the most common congenital infection in the United States affecting 0.3% of non-Hispanic white infants, with some data suggesting higher prevalence rates in black and multiracial infants [1,2]. As such, cCMV is the leading infectious cause of birth defects and non-genetic hearing loss [3,4]. Of cCMV-infected children identified through universal screening, approximately half of the symptomatic children and 13.5% of asymptomatic children experienced long-term health problems—most notably sensorineural hearing loss (SNHL) [5].

cCMV screening practices are inconsistent across the United States due to absent or inadequate clinical standards. However, cCMV diagnosis is a prerequisite for optimal evaluation and management of all infected infants [6]. Confirming that the virus was acquired prenatally is critical, as postnatal infection rarely causes significant disease. Therefore, testing must be performed within 3 weeks of age, after which the opportunity to diagnose congenital infection is nearly lost. Historically, testing was only performed on so-called “symptomatic” infants with visible or laboratory evidence of cytomegalic inclusion disease, such as microcephaly, hepatosplenomegaly, thrombocytopenia, or purpuric rash. When cCMV occurs without visible signs at birth, but with the potential for hearing loss and progressive disease it is described as “asymptomatic cCMV”. As awareness of this entity has grown, some states or hospitals have adopted “targeted” screening for infants who do not pass the newborn hearing screen [7]. Some long-term effects of moderate-to-severe symptomatic disease can be mitigated if treatment with oral valganciclovir is initiated within 4 weeks of age [8]. However, without cCMV diagnosis to prompt further evaluation for those effects, many infected infants will not be considered for this treatment. Moreover, all infected infants regardless of disease severity benefit from a consistent monitoring schedule to provide early detection of hearing loss and other developmental delays [6].

The Massachusetts cCMV Coalition (MCC; https://cmvmass.org) was established in January 2019 as an organization of family members, physicians, audiologists, public health officials, educators and other stakeholders to advance local and national efforts to understand, prevent, identify, and treat cCMV infection. While the Commonwealth of Massachusetts currently screens all newborns for 66 different conditions, cCMV is not among them. To inform any future development of state-sponsored testing guidelines, the MCC designed a survey to assess the current state of neonatal CMV screening at all birth hospitals in Massachusetts.

## 2. Materials and Methods

The study was approved by the Institutional Review Board of Massachusetts Eye and Ear, Mass General Brigham, protocol number 2020P002951. Contact information was collected between November 2020 and February 2021 for all 45 active newborn nurseries in Massachusetts to identify potential survey respondents, including newborn hearing screening program directors, nurse managers, and neonatology professionals. Invitation letters were sent by email describing the rationale for the study and asking them to complete a survey through REDCap, a secure electronic data capture system hosted at Mass Eye and Ear, Mass General Brigham. To maximize engagement, emails were redistributed periodically to non-responding centers and followed up with direct phone calls and additional emails. When respondents reported that they were unable to answer some or all of the survey questions themselves, they were encouraged to forward the survey to colleagues who were more familiar with cCMV testing or clinical protocols at their institutions. Some responses were modified by respondents after further research into hospital practices. In sites with multiple respondents, we collaborated with the sites to collate answers into a single form per site to eliminate any duplication of data. Respondents were given approximately two months to complete the survey.

Questions were designed collaboratively by members of the MCC based on their areas of expertise. All survey items used a checkmark system that allowed respondents to select multiple answers per question so that the sum of responses often exceeded the number of respondents. In addition, many of the questions included the option to choose “other”, which gave respondents space to provide comments.

Descriptive statistics were generated after exporting the survey data to Microsoft Excel (Microsoft, Redmond, WA, USA).

## 3. Results

We successfully communicated by email with program directors of universal newborn hearing screening programs at all 45 Massachusetts active newborn nurseries. Following these initial communications, representatives of 33 hospitals (73.3%) completed at least one question on the survey.

### 3.1. Characteristics of Responding Hospitals

Table 1 describes the characteristics of Massachusetts birth hospitals. Most of the 33 responding hospitals provided services in general pediatrics (72.7%), neonatology (66.7%), and audiology (54.5%), but fewer than half provided services in pediatric infectious disease (18.2%) or otolaryngology (33.3%). Birth centers with a neonatal intensive care unit (NICU) were significantly more likely to respond than those without a NICU (Table 2, *p* = 0.02, Fisher’s exact test).

### 3.2. Newborn Hearing Screening Protocols

Newborn hearing screening is typically performed by a nurse, technician, and/or audiologist in 20 (60.6%), 16 (48.5%), and 14 (42.4%) of the responding hospitals, respectively. 9 (27.3%) hospitals use an external screening service or other professional. The total was greater than 100% as screening is performed by more than one type of professional in some centers. The rate of referral (failure) on screening was <3% at 21 (63.6%) and 3–5% at 8 (24.2%) hospitals (Table 3).

### 3.3. cCMV Screening Practices

Only 16 hospitals (48.5% of respondents) reported having any established approach to cCMV screening within 3 weeks of age, while 8 (24.2%) had no defined plan and 9 (27.3%) did not know or did not answer (Table 4). Of the 16 hospitals with a cCMV screening approach, 9 (56.3%) reported having a written protocol approved by a relevant committee, provider group, and/or unit. 15 hospitals with an established approach to CMV (93.8%) screen infants who refer in one or both ears on the hearing screen, but few report that they screen all infants admitted to the nursery (1 or 6.3%) or to the NICU (2 or 12.5%). Twelve hospitals (36.4% of total respondents) reported having providers who performed screening without a written protocol.

### 3.4. cCMV Screening Modalities

Large fractions of responding hospitals did not answer detailed questions about cCMV screening modalities (Table 5). Of responders, 15 (45.5%) collected saliva samples for cCMV screening, while 8 (24.2%) and 2 (6.1%) obtained urine or blood, respectively. Polymerase chain reaction (PCR) testing was the most common assay used among 11 (33.3%) respondents, while 2 (6.1%) used viral culture. Only 5 (15.2%) hospitals routinely performed a confirmatory assay when the initial screening test was positive, with urine PCR or urine culture most commonly used (Table 5).

### 3.5. Communication of Positive CMV Test Results

Most (29 or 87.9%) respondents did not know or did not answer how results of cCMV screening were reported to families and who was responsible for coordinating care for cCMV-infected infants. The most common reported communication plans involved the lab notifying ordering providers (5 or 15.2%) and the primary care physician contacting families with results (4 or 12.1%) and coordinating subsequent care (5 or 15.2%) (Table 6).

### 3.6. Plans to Develop an Approach to cCMV Screening

Most (28 or 84.8%) respondents did not know, did not answer, or had no plan for developing a specific approach to cCMV screening, and only 6 (18.2%) are in various stages of developing or expanding protocols. One (3.0%) birth hospital reported that a plan for developing an approach was in place (Table 7).

## 4. Discussion

The results of this survey indicate that fewer than half of Massachusetts birth hospitals are consistently screening infants for cCMV infection. Of those who report a screening approach, few have a written protocol that is vetted and practiced by relevant care providers. Only two hospitals perform cCMV screening on all infants admitted to the newborn nursery or NICU. While the overall survey response rate was high, a large proportion of respondents did not know the answers to all questions, suggesting limited multi-disciplinary collaboration or awareness of cCMV screening practices. Of note, response to the survey was significantly higher for hospitals that provide neonatal intensive care, possibly due to more frequent experience evaluating and caring for infants with severe cCMV infection.

Many reports have outlined the benefits of universal cCMV screening and early detection [4,6,9,10,11,12,13]. Some newborns with cCMV infection may be candidates for early antiviral therapy, including those with no visible abnormalities at birth but for whom evaluation prompted by confirmed infection (e.g., laboratory studies, head imaging, and audiologic assessment) show that therapy is indicated. In one retrospective study, oral valganciclovir therapy was shown to restore normal hearing function in about two-thirds of affected ears for infants with isolated hearing loss [14,15]. Moreover, antiviral therapy after 4 weeks of age may benefit infants who are asymptomatic at birth but experience delayed-onset hearing loss [16,17,18].

Despite the lack of evidence to clarify some areas of cCMV management—especially for patients with no detectable abnormalities at birth—cCMV diagnosis allows anticipatory guidance for families and monitoring for late onset sequelae especially hearing loss.

The overall lack of awareness among pregnant women, families, the lay public, clinicians, and policymakers contributes to high cCMV birth prevalence. To address this deficit, advocates in many states have pursued legal mechanisms for CMV education and/or newborn testing. Recently, Minnesota then New Jersey were the first to pass legislation mandating universal newborn CMV screening. Minnesota’s “Vivian Act” was championed by the family of 7-year-old Vivian Henrikson with cCMV infection and passed into law in June 2021 with bipartisan support. This legislation supports statewide outreach education programs to promote awareness of cCMV, and as of early 2022, funding for universal screening. New Jersey’s bipartisan bill mandating a maternal CMV education program and universal newborn cCMV screening was signed by Governor Murphy in January 2022. Of note, per this legislation universal newborn cCMV screening will not be initiated until it is included in the Recommended Uniform Screening Panel (RUSP) of the United States Department of Health and Human Services.

Similarly in February 2021, the Massachusetts CMV Coalition (MCC) participated in drafting and filing a bill that would mandate prenatal education and universal screening of all newborns in Massachusetts. Indiana and Kentucky have also filed bills with their state legislatures requiring universal or targeted screening. Legislation is currently pending in Pennsylvania (education and targeted screening) and Michigan (education). Instead of taking legislative action, California has established a multi-disciplinary committee to investigate the best approach to cCMV education and testing in the state.

Several states have already enacted cCMV legislation. Florida recently passed SB 292 on (targeted screening; https://www.flsenate.gov/Session/Bill/2022/292 (accessed on 6 April 2022). Similarly, the Maine senate unanimously passed Bill LD 1747 (targeted screening; https://legislature.maine.gov/legis/bills/bills_128th/billtexts/SP003601.asp (accessed on 25 April 2022), thereby routing the bill to the governor for signature. According to the National CMV Foundation (https://www.nationalcmv.org (accessed on 25 April 2022)), several other states have already instituted education and/or testing policies. Utah, Illinois, Iowa, and New York have mandated both prenatal education and targeted cCMV screening, while Connecticut and Virginia require targeted screening only. Colorado, Hawaii, Idaho, Illinois, Iowa, New York, Oregon, Texas, Minnesota and Utah require cCMV awareness education for the general public and health professionals, while Tennessee mandates education for women of childbearing age only.

Several studies have shown that regardless of the approach, cCMV screening is cost effective. Bergevin et al. [19] examined the cost effectiveness of the cCMV education and targeted screening program established by Utah, which in 2013 became the first state in the U.S. to mandate cCMV screening through legislation. The authors concluded that the advantages of the program outweighed the costs based on the assumption that initiation of antiviral treatment would likely benefit hearing levels for at least one year. This effect could not only delay purchase of expensive hearing technology, but also provide children with improved access to speech sounds during critical speech/language learning periods, thus reducing the need for special education in the future. Furthermore, mandatory educational programs were predicted to decrease the incidence of cCMV, resulting in significant reduction in public spending [20]. Similarly, Gantt et al. provided a comprehensive model of cost effectiveness for both targeted and universal cCMV screening programs. While both were found to have overall cost benefit, they favored universal screening to yield greater cost benefit and to allow directed care for all infected infants. Specifically, such a model would include infants without apparent symptoms or hearing loss at birth who might otherwise be lost to follow-up [20,21]. In a review of published economic analyses of cCMV detection and prevention efforts, Grosse et al. [22] found that these and other studies reported favorable cost effectiveness of the interventions examined, and that data from new CMV screening programs implemented over time will continue to inform policymakers and legislation.

## 5. Conclusions

Although cCMV is the most common congenital infection in the United States, awareness of preventative measures and clinical effects in children is limited [12]. Our survey results confirm that like many other states, Massachusetts hospital cCMV screening protocols are absent or inconsistent. While our study focused on Massachusetts hospitals, there is nothing to suggest that Massachusetts is unique in this case. A cohesive strategy involving standardized education and screening guidelines is needed to reduce the incidence and burden of cCMV disease on children and their families, and a legislative mandate may be an effective way to achieve this goal. Such a strategy may be considered at the state and federal level.

## Figures and Tables

**Table 1 IJNS-08-00065-t001:** Responses to demographic questions.

Question	Response	*N*	Proportion	
			of Centers Queried (45)	of Centers Responding (33)
Type(s) of on-site pediatric clinical providers:	General Pediatrics	24	53.3%	72.7%
	Neonatology	22	48.9%	66.7%
	Audiology	18	40.0%	54.5%
	Otolaryngology	11	24.4%	33.3%
	Pediatric Emergency	11	24.4%	33.3%
	Pediatric Infectious Disease	6	13.3%	18.2%
	Other ^1^	5	11.1%	15.1%
	Unknown	0	0.0%	0.0%
	Not Answered	0	n/a	0.0%

^1^ Other responses included: Midwives (2), Family Medicine (2), and Pediatric Hospitalists (1).

**Table 2 IJNS-08-00065-t002:** Contingency table of response rates against maximum level of care as defined by the Code of Massachusetts Regulations (CMR).

	Level 1 or 1.5 : Well-Baby Nursery or Continuing Care Nursery	Level 2 or 2.5: Special Care Nursery	Level 3 : NICU	Totals
Responded	13	9	11	33
Did not respond	4	8	0	12
Totals	17	17	11	45
Response rate	76.5%	52.9%	100.0%	73.3%

Fisher’s exact test, *p* = 0.02.

**Table 3 IJNS-08-00065-t003:** Responses regarding newborn hearing screening practices and outcomes.

Question	Response	*N*	Proportion	
			of Centers Queried (45)	of Centers Responding (33)
Type(s) of hearing screeners:	Nurse	20	44.4%	60.6%
	Audiologist	14	31.1%	42.4%
	External Screening Service	4	8.9%	12.1%
	Technician	16	35.6%	48.5%
	Other ^2^	5	11.1%	15.2%
	Unknown	0	0.0%	0.0%
	Not Answered	0	0.0%	0.0%
Estimated annual hearing screen refer rate:	<3%	21	46.7%	63.6%
	3–5%	8	17.8%	24.2%
	>5%	0	0.0%	0.0%
	Exact Refer Rate Known ^3^	7	15.6%	21.2%
	Unknown	3	6.7%	9.1%
	Not Answered	1	n/a	3.0%

^2^ Other responses included: Audiology Interns or Assistants (3), Certified Nursing Assistants (1), and Co-Op Students (1). ^3^ Exact refer rates reported were: 0% (1), 1.4% (1), 1.89% (1), 2.3% (1), 2.5% (1), 3.5% (1), and 4.19% (1).

**Table 4 IJNS-08-00065-t004:** Responses regarding congenital cytomegalovirus screening approaches.

Question	Response	*N*	Proportion	
			of Centers Queried (45)	of Centers Responding (33)
Does your birth center have a specific approach to cCMV screening within 3 weeks of age? (*Please answer based on status before the COVID-19 pandemic*)	Yes	16	35.6%	48.5%
	No	8	17.8%	24.2%
	Unknown	3	6.7%	9.1%
	Not Answered	6	13.3%	18.2%
Type(s) of specific approaches to cCMV screening within 3 weeks of age:	Written protocol approved by institutional committee	2	4.4%	6.1%
	Written protocol approved by relevant providers (e.g., General Pediatrics, Infectious Disease, and/or Neonatology)	3	6.7%	9.1%
	Written protocol approved by relevant unit (e.g., Nursery, NICU, other advanced care unit)	4	8.9%	12.1%
	Common practice for some/all providers or units but no written protocol	12	26.7%	36.4%
	Other	0	0.0%	0.0%
	Unknown	3	6.7%	9.1%
	Not Answered	16	n/a	48.5%
What neonatal population(s) is/are screened?	All admitted to the nursery (universal)	1	2.2%	3.0%
	All admitted to the NICU or other advanced care unit (universal)	2	4.4%	6.1%
	Newborn hearing screen refer (one or both ears)	15	33.3%	45.5%
	All born to women with human immunodeficiency virus infection	1	2.2%	3.0%
	All born to women with Hepatitis C infection	0	0.0%	0.0%
	All born with other identified CMV risk factors	13	28.9%	39.4%
	Other ^4^	2	4.4%	6.1%
	Unknown	3	6.7%	9.1%
	Not Answered	13	n/a	39.4%

^4^ Other responses included: <34 weeks gestational age (1), Low Birth Weight (1).

**Table 5 IJNS-08-00065-t005:** Responses regarding congenital cytomegalovirus screening modalities.

Question	Response	*N*	Proportion	
			of Centers Queried (45)	of Centers Responding (33)
What sample type is used for screening?	Saliva	15	33.3%	45.5%
	Urine	8	17.8%	24.2%
	Blood	2	4.4%	6.1%
	Other	0	0.0%	0.0%
	Unknown	3	6.7%	9.1%
	Not Answered	14	31.1%	42.4%
What assay is used for screening?	PCR	11	24.4%	33.3%
	Culture	2	4.4%	6.1%
	Other	0	0.0%	0.0%
	Unknown	9	20.0%	27.3%
	Not Answered	14	n/a	42.4%
What type of laboratory performs the screening test?	Clinical reference lab	7	15.6%	21.2%
	Internal or affiliated hospital lab	3	6.7%	9.1%
	Research lab	1	2.2%	3.0%
	Other ^5^	1	2.2%	3.0%
	Unknown	11	24.4%	33.3%
	Not Answered	14	n/a	42.4%
If a screening CMV test is positive, is a confirmatory test routinely performed?	Yes	5	11.1%	15.2%
	No	6	13.3%	18.2%
	Unknown	12	26.7%	36.4%
	Not Answered	12	n/a	36.4%
What sample type is used for confirmation?	Saliva	0	0.0%	0.0%
	Urine	4	8.9%	12.1%
	Blood	1	2.2%	3.0%
	Other	0	0.0%	0.0%
	Unknown	12	26.7%	36.4%
	Not Answered	19	n/a	57.6%
What assay is used for confirmation?	PCR	3	6.7%	9.1%
	Culture	3	6.7%	9.1%
	Other	0	0.0%	0.0%
	Unknown	12	26.7%	36.4%
	Not Answered	19	n/a	57.6%
What type of laboratory performs the confirmatory test?	Clinical reference lab	2	4.4%	6.1%
	Internal or affiliated hospital lab	1	2.2%	3.0%
	Research lab	0	0.0%	0.0%
	Other ^6^	1	2.2%	3.0%
	Unknown	12	26.7%	36.4%
	Not Answered	19	n/a	57.6%

^5^ other responses included: External Institution/Hospital (1). ^6^ other responses included: External Institution/Hospital (1).

**Table 6 IJNS-08-00065-t006:** Responses regarding communication of cCMV screening results.

Question	Response	*N*	Proportion	
			of Centers Queried (45)	of Centers Responding (33)
If a confirmatory CMV test is positive, what types of communication and follow-up strategies are used?	Lab notifies ordering provider or unit	5	11.1%	15.2%
	Lab notifies primary care physician	2	4.4%	6.1%
	Lab notifies specialist	2	4.4%	6.1%
	Ordering provider or unit notifies family	1	2.2%	3.0%
	Primary care physician notifies family	5	11.1%	15.2%
	Specialist notifies family	2	4.4%	6.1%
	Ordering provider or unit coordinates referral	3	6.7%	9.1%
	Primary care physician coordinates referral	4	8.9%	12.1%
	Specialist coordinates referral	2	4.4%	6.1%
	Other	0	0.0%	0.0%
	Unknown	11	24.4%	33.3%
	Not Answered	18	n/a	54.5%

**Table 7 IJNS-08-00065-t007:** Responses regarding plans for a specific approach to cCMV screening.

Question	Response	*N*	Proportion	
			of Centers Queried (45)	of Centers Responding (33)
What is the plan for developing a specific approach to cCMV screening at your center?	No plan at this time	1	2.2%	3.0%
	Developing an approach is being considered	2	4.4%	6.1%
	An approach is currently being developed	1	2.2%	3.0%
	Our facility is investigating or planning to expand the current protocol	3	6.7%	9.1%
	Plan already in place	1	2.2%	3.0%
	Unknown	9	20.0%	27.3%
	Not Answered	18	n/a	54.5%

## Data Availability

Data supporting reported results are available upon request to the corresponding author.

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
