# Peer review of "Congenital Cytomegalovirus Screening in Massachusetts Birth Hospitals: A Statewide Survey"

_2409-515X, 2022, doi:10.3390/ijns8040065_

Round 1
Reviewer 1 Report
I read with great interest your manuscript of cCMV infection and hearing loss. In a retrospective study of an expert ENT center in Belgium it was shown that Cochlear nerve deficiency and congenital cytomegalovirus infection accounted for the majority of unilateral hearing loss in newborns (Boudewyns A et al); in the past we published with A. Boudewyns a paper where the efficacy of detecting CMV in DBS/newborn screening card by qPCR was proven in newborns and in children who manifested hearing loss at a later age; so this is also a possible approach by confirming congenital CMV in DBS rather than in saliva and urine (excretion is limited in time).
I totally agree with you that targeted cCMV screening should be performed in every newborn with abnormal hearing screening test, and perhaps by universal screening for cCMV by qPCR to detect also the children with late-onset hearing loss; a structured referral to and follow-up in an ENT unit with expertise in this matter is recommended.
Author Response
The authors thank the reviewer for this favorable report and agree with the reviewer's thoughts on other modalities for CMV testing.
Reviewer 2 Report
This study reports the results of a survey that was administered to Massachusetts hospitals regarding protocols for congenital CMV testing and follow-up. The authors found that fewer than half of hospitals have an established protocol for CMV screening, and even fewer reported coordination of care for children with positive screen. The authors highlight this lack of standardization as evidence to support improved education and screening guidelines for congenital CMV. These results add to the growing evidence of a gap in diagnostic testing and care delivery for children with congenital CMV.
Introduction:
1) The sentence beginning “As awareness of the impact…” (lines 52-55) is difficult to follow as written. Please revise. Consider separating into two sentences.
2) The final paragraph seems to narrow the scope of this paper to informing public health policy in Massachusetts. Could the authors comment on the generalizability of these results? How would they be relevant to a wider scientific community?
Materials and Methods
1) The method of dissemination of the survey seems somewhat haphazard. What was the method of quality control to ensure that the authors had contacted the correct respondent(s)?
2) Did each newborn nursery have only one survey form available in REDCap, or could there be more than one? Was there the potential for multiple respondents from a single newborn nursery?
3) The authors have published the survey results in Table format, which makes it more difficult to delve deeper into the results. Did the authors investigate what characteristics made certain hospitals more likely to have protocol? Is there additional analysis that can be done by looking at the characteristics of the hospitals?
Results
1) Could the authors clarify what they mean by “reached potential respondents”?
2) I would recommend revising the sentence lines 100-102 as follows: “Birth centers with a neonatal intensive care unit (NICU) were significantly more likely to respond than those without a NICU.”
3) Tables. I do not think it is accurate to report a proportion for the “centers queried” if some of the centers did not respond? This is not a true proportion due to missing data, and it is confusing to the reader.
4) Table 1. It may be easier for the reader if the specialties were reported in descending order (ie moving from higher N to lower N).
5) Is it correct for the table to include “not answered” = 0 for “proportion of centers queried” if some of these centers did not respond?
6) Please revise lines 120-123 as it is difficult to follow. Consider separating into two or three sentences.
Discussion
1) Lines 187-188: The authors mention that a “large proportion of audiologists or neonatology professionals did not…” I was curious why these two specialties were highlighted here? Did the authors evaluate respondents’ knowledge by specialty?
2) I would ask the authors to note that New Jersey’s law calls for implementation of universal screening only after it is incorporated into the RUSP at the national level (https://legiscan.com/NJ/text/S3975/2020). Please review and add details as needed.
3) Please revise lines 248-252. This sentence is difficult to follow
Author Response
- The sentence beginning “As awareness of the impact…” (lines 52-55) is difficult to follow as written. Please revise. Consider separating into two sentences.
Thank you for this comment. The manuscript has been clarified as follows:
When cCMV occurs without visible signs at birth, but with the potential for hearing loss and progressive disease it is described as “asymptomatic cCMV”. As awareness of this entity has grown, some states or hospitals have adopted “targeted” screening for infants who do not pass the newborn hearing screen
- The final paragraph seems to narrow the scope of this paper to informing public health policy in Massachusetts. Could the authors comment on the generalizability of these results? How would they be relevant to a wider scientific community?
We agree that these findings are likely to be generalizable, however did not wish to make conclusions that extended beyond the data in our study. We have added the following comments to the paragraph in question:
While our study focused on Massachusetts hospitals, there is nothing to suggest that Massachusetts is unique in this case. A cohesive strategy involving standardized education and screening guidelines is needed to reduce the incidence and burden of cCMV disease on children and their families, and a legislative mandate may be an effective way to achieve this goal. Such a strategy may be considered at the state and federal level.
Materials and Methods
- The method of dissemination of the survey seems somewhat haphazard. What was the method of quality control to ensure that the authors had contacted the correct respondent(s)?
As screening practices are markedly heterogeneous among birth hospitals, we were required to use a flexible strategy in order to maximize study participation. We began with listed contacts for newborn nurseries and distributed the survey to all relevant staff at each institution. For institutions that did not respond completely, we requested contact information for additional staff who may be able to complete survey information. This process was repeated exhaustively with emails and with phone calls. Despite these efforts quite a number of programs did not respond.
- Did each newborn nursery have only one survey form available in REDCap, or could there be more than one? Was there the potential for multiple respondents from a single newborn nursery?
Multiple participants were allowed to participate from each site in order to get the most complete data set; In sites with multiple respondents, we collaborated with the sites to collate answers into a single form per site to eliminate any duplication of data. The manuscript has been edited to include the following:
In sites with multiple respondents, we collaborated with the sites to collate answers into a single form per site to eliminate any duplication of data.
- The authors have published the survey results in Table format, which makes it more difficult to delve deeper into the results. Did the authors investigate what characteristics made certain hospitals more likely to have protocol? Is there additional analysis that can be done by looking at the characteristics of the hospitals?
We did do some analysis by level of care provided at each birth center. Specifically:
Birth centers with a neonatal intensive care unit (NICU) were significantly more likely to respond than those with lower levels of care (Table 2, p = 0.02, Fisher’s exact test).
Due to the small sample size it was found that additional statistics based on hospital type would not be meaningful.
Results
- Could the authors clarify what they mean by “reached potential respondents”?
This has been edited to read:
We successfully communicated by email with program directors of universal newborn hearing screening programs at all 45 Massachusetts active newborn nurseries. Following these initial communications, representatives of 33 hospitals (73.3%) completed at least one question on the survey.
- I would recommend revising the sentence lines 100-102 as follows: “Birth centers with a neonatal intensive care unit (NICU) were significantly more likely to respond than those without a NICU.”
Thank you. This has been changed to read, “Birth centers with a neonatal intensive care unit (NICU) were significantly more likely to respond than those without a NICU (Table 2, p = 0.02, Fisher’s exact test).”
- I do not think it is accurate to report a proportion for the “centers queried” if some of the centers did not respond? This is not a true proportion due to missing data, and it is confusing to the reader.
Thank you for this comment. We did consider leaving out this column, however given that there is some information to be drawn by considering the total number of hospitals queried, we did decide to leave this in place in order to allow the readers to have this information.
- Table 1. It may be easier for the reader if the specialties were reported in descending order (ie moving from higher N to lower N).
Thank you. This change has been made in Table 1
- Is it correct for the table to include “not answered” = 0 for “proportion of centers queried” if some of these centers did not respond?
By including the alternate “n” of total hospitals queried, we acknowledge that the inclusion of this calculation is imperfect, but as above does help the reader consider responses in the context of this larger number. As such, this cell applies that larger N to the number of respondents who didn’t answer that question. This is admittedly imperfect, and as such for this cell, the value has been replaced with “n/a” wherever it appears in tables throughout.
- Please revise lines 120-123 as it is difficult to follow. Consider separating into two or three sentences.
Thank you for this comment. The text has been revised to read:
Newborn hearing screening is typically performed by a nurse, technician, and/or audiologist in 20 (60.6%), 16 (48.5%), and 14 (42.4%) of the responding hospitals, respectively. 9 (27.3%) hospitals use an external screening service or other professional. The total was greater than 100% as screening is performed by more than one type of professional in some centers. The rate of referral (failure) on screening was < 3% at 21 (63.6%) and 3–5% at 8 (24.2%) hospitals (Table 3).
Discussion
1) Lines 187-188: The authors mention that a “large proportion of audiologists or neonatology professionals did not…” I was curious why these two specialties were highlighted here? Did the authors evaluate respondents’ knowledge by specialty?
Thank you for this comment. The reviewer is correct that we did not evaluate knowledge by specialty and that this was confusing. We have revised the text as follows:
While the overall survey response rate was high, a large proportion of respondents did not know the answers to all questions, suggesting limited multi-disciplinary collaboration or awareness of cCMV screening practices.
2) I would ask the authors to note that New Jersey’s law calls for implementation of universal screening only after it is incorporated into the RUSP at the national level (https://legiscan.com/NJ/text/S3975/2020). Please review and add details as needed.
Thank you for this point. The text has been amended as follows:
Of note, per this legislation universal newborn cCMV screening will not be initiated until it is included in the Recommended Uniform Screening Panel (RUSP) of the United States Department of Health and Human Services.
3) Please revise lines 248-252. This sentence is difficult to follow
Thank you for your comment. The sentence has been revised as follows:
While both were found to have overall cost benefit, they favored universal screening to yield greater cost benefit and to allow directed care for all infected infants. Specifically, such a model would include infants without apparent symptoms or hearing loss at birth who might otherwise be lost to follow-up [21, 22].
Round 2
Reviewer 2 Report
I appreciate the authors' detailed responses to the suggested edits. This paper will add to the literature regarding screening efforts for congenital CMV.